# Slacktivity: Scaling Slack for Large Organizations

Authors Anonymous

For Submission

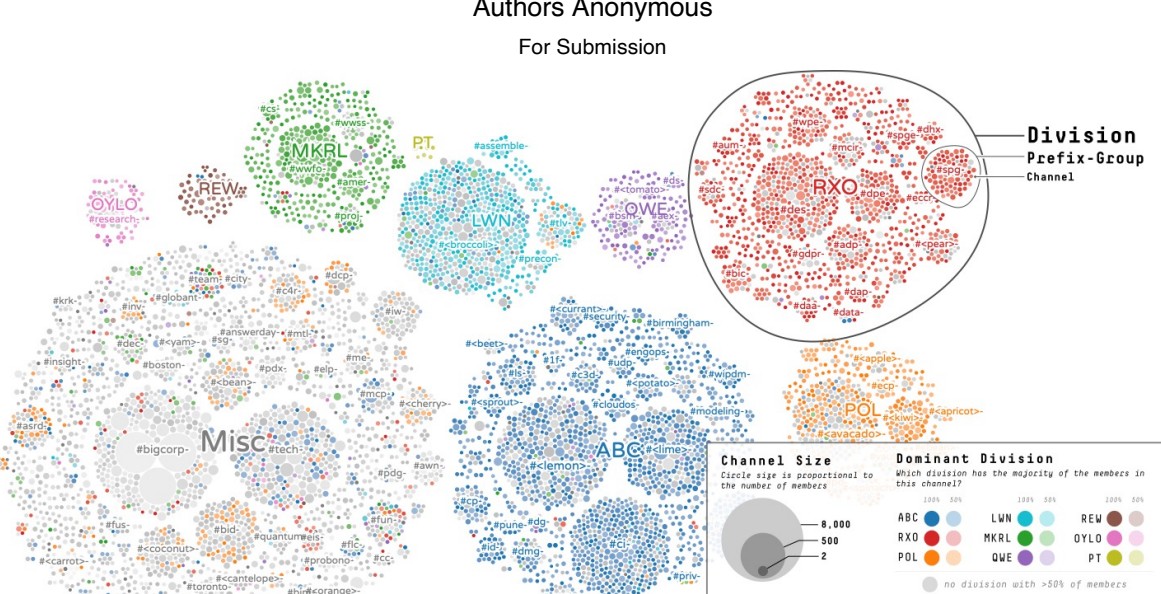

Figure 1: Slacktivity's galaxy view. Each circle in the visualization is a Slack channel. (Note: division names have been changed to remove identifying information, and select channel names have been replaced with <fruit|vegetable>.)

## ABSTRACT

Group chat programs, such as Slack, are a promising and increasingly popular tool for improving communication among collections of people. However, it is unclear how the current design of group chat applications scales to support large and distributed organizations. We present a case study of a company-wide Slack installation in a large organization (>10,000 employees) incorporating data from semi-structured interviews, exploratory use, and analysis of the data from the Slack workspace itself. Our case study reveals emergent behaviour, issues with exploring the content of such a large Slack workspace, and the inability to keep up with news from across the organization. To address these issues, we designed Slacktivity, a novel visualization system to augment the use of Slack in large organizations and demonstrate how Slacktivity can be used to overcome many of the challenges found when using Slack at scale.

**Keywords**: Group chat, visualization.

## 1 INTRODUCTION

As organizations become larger and more distributed, communication becomes increasingly difficult. Even when workers are co-located, technologies like email and instant messaging are frequently used as an alternative to direct face-to-face communication. Email is often thought of as asynchronous and useful for long-term searching, while instant messaging is useful for synchronous communication and quicker conversations. More recently, group chat systems (such as Slack [1] and Microsoft Teams [2]) have become popular, combining benefits of both email and instant messaging, while offering the potential for increased collaboration and coordination.

When used over a period of time, a group chat workspace has the potential to be a useful "knowledge base" for a company, storing a history of communication about a given topic. However, the design of current group chat systems is optimized for near real-time communication, making it difficult to derive insights from this wealth of historical information [3]. This problem is exacerbated for large organizations using Slack where there may be thousands of channels and millions of messages. Additionally, with so many channels and messages, it is neither feasible nor possible to keep up with all activity across all channels to get a sense of what is going on throughout the organization, and the task of figuring out which channels to join or post questions to becomes more difficult.

In this paper we present a case study of the internal usage of Slack at BigCorp,[1] a ten thousand person software company which has been using a unified Slack workspace for the past 3+ years, with over 15,000 channels created and 65 million messages sent over that time. Our case study combines exploratory use of the Slack workspace, formal and informal interviews, and data analysis of the use of Slack by BigCorp. Using this data, we identified strategies employees use to cope with Slack at scale and also the pain points of using Slack such as the inability to find historical information, and the need to keep up with their organization better.

Using the findings from our case study we present Slacktivity, a tool to address the limitations of Slack when being used within a large organization. Slacktivity gives an overview of the channels across the entire organization with a cluster view, and also allows for detailed exploration of the entire history of a channel.

This paper makes two main contributions: a case study to better understand how group chat is used in large organizations, and a novel interactive visualization system to augment the use of Slack in a workplace.

---

[1] Name anonymised for submission.

## 2 BACKGROUND AND RELATED WORK

### 2.1 Introduction to Group Chat

Group chat works by combining the functionality of instant messaging and Internet Relay Chat (IRC) with rich communication. The most basic form of communication in group chat is a message. Messages can contain text, images or other attachments and can reply to one another, forming a thread. However, threads in group chat are typically limited to one level of replies. In addition to replying to a message, a user can also react. Reactions are small emoji-like glyphs that are counted just below the message. When responding with a reaction there is no notification sent, making this a non-intrusive form of communication. Messages can be sent to three different locations: direct messages, private channels, and public channels. Direct messages can be sent to groups of up to 8 people. Channels on the other hand can hold any number of people and are identified by a channel name (eg. #general). Channels can be either public, meaning anybody can join, or private, which require an invitation. Finally, all channels and direct messages are part of a workspace, the highest level of hierarchy in group chat.

Our work explores the use of group chat in the workplace, and the use of visualization to improve its usage. This work looks at, and refers to Slack as the group-chat system, but the concepts and ideas apply equally to other systems (such as Microsoft Teams [2]).

### 2.2 Communication in the Workplace

Technology plays a key role in supporting communication in the workplace, but its use is varied and complicated. When face to face interaction is needed but not possible, video conference software like Skype or Google Hangouts is used. However, most communication does not require such an expressive communication style and can instead be textual. Recently, instant messaging has been a popular communication tool in the workplace [4], [5]. A case study [5] of a large tech company from 2005 showed that 38% of communication took place over instant messaging, nearly equal to the 39% with email. Only 23% of communication was face-to-face or using the phone. Email is also used extensively and many people view their email as more than a communication tool, as a "personal information store" [6].

Another aspect of communication in the workplace is social media. Traditional non-workplace social medias like Facebook and Twitter have been demonstrated to be useful to organizations through the creation of weak ties [7]. Furthermore, some social networks have been designed specifically for use in the workplace. WaterCooler [8] was designed for use at HP to aggregate various content from across the organization. Another more commercialised approach to social networking is the Yammer tool, which is aimed at bringing a social network more comparable to Facebook into the workplace.

In recent years group chat has been used increasingly in the workplace, beginning with the inception of Slack in 2013 which had over 10 million active daily users in 2019 [9]. Despite the rapid adoption of group chat in the workplace, Zhang and Cranshaw [10] identify several issues associated with group. They report that employees struggle to find old information; chat history is overwhelming to newcomers; and employees fail to keep up with multiple channels. Unlike email, it is unclear if group chat is also used as a "personal information store".

Communication tools have the opportunity to be a critical part of an organization's knowledge management strategy as critical discussions often occur over email, instant messaging, or group chat. Research has shown how an organization can use instant messaging [11], [12] and email [13] as a part of their knowledge management strategy, but it is unknown how group chat can fit in.

### 2.3 Visualization of Conversation

Conversation exists in many mediums, such as forums, emails and instant messaging, and the academic literature has explored many different ways to visualize it either during the creation of the conversation, or after the conversation has occurred. Pioneering work began with the visualisation of forums, with a special focus on Usenet. Smith and Fiore [14] visualized Usenet threads by displaying the structure of the thread as a tree augmented with information about the users involved in the thread. Turner et. al [15] visualize Usenet as a hierarchical strategy to make recommendations on how to cultivate and manage Usenet groups. Wikum [16] uses recursive summarisation and visualisation to make the overcome the scale of online discussions. Other techniques have explored visualizing conversations by taking advantage of threads, including Thread Arcs [17], ThemeRiver [18], tldr [19], iForum [20], and Newman's work [21].

Venolia and Neustaedter [22] visualized email by creating conversations from the sequence and reply relationships, building on the idea of threading. Conversation thumbnails [23] visualize each email in a conversation as a rectangle, displayed in the order they were received and representing the complexity of the conversation.

Bergstrom and Karahalios [24] designed Conversation Clusters as a method of archiving instant message conversations in a manner that is easy to retrieve a desired conversation. Conversation Clusters visualize the groups of salient words by using colour. However, most work in instant messaging has looked at augmenting communication by changing the interface you interact with, such as giving people movable circles for their avatars [25], [26] or using a visualization to foster positive behavioural changes [27]. Our work differs from prior work visualising conversation in that we augment each mark with more information from the chat and also break the sequential linear relationship with time.

### 2.4 Understanding Group Chat

Efforts to understand group chat better are relatively rare. Our work is most related to T-Cal [28], which visualizes Slack conversations using a calendar-based visualization and allows for in-depth exploration using their thread-pulse design. However, T-Cal was designed to address the needs of Slack workspaces for massively open online courses (MOOCs) that are only used for a set amount of time. Another approach was taken by Zhang and Cranshaw [10], who aimed to improve sensemaking of group chat by creating a Slack bot, Tilda, that assisted in collaborative tagging of conversations. Both our case study of Slack and Slacktivity build on and draw inspiration from T-Cal and Tilda, however our focus is discovering and addressing the particular problems faced by deploying Slack at a large scale.

## 3 CASE STUDY: SLACK AT BIGCORP

To get a better understanding of how Slack is used "at scale", we studied Slack usage at *BigCorp* – a multinational design software company with over 10,000 employees distributed at dozens of locations around the world. BigCorp has officially adopted and encouraged the use of Slack as a platform for group chat. Further, BigCorp has consolidated Slack activity into a single unified Slack workspace, rather than allowing teams to maintain their own individual Slack workspaces.

### 3.1 Methodology

Our case study incorporates data from several sources: aggregate analysis from an export of the entire (non-private) history of the Slack workspace, extensive exploratory analysis, an interview with the Director in charge of Slack at BigCorp, informal discussions with dozens of employees about their Slack usage, and formal interviews with 5 employees (3 heavy users of Slack, 1 occasional user, and 1 infrequent user – referred to as P1-P5).

Our aggregate data analysis uses all Slack messages sent on all internally-public channels (that is, channels that all employees of BigCorp can see). This includes all of the data about threads and reactions. Every public channel is included and its metadata such as descriptions and creation time. We analyse this data using only simple summary statistics and visualisation. We also combined the Slack user profiles with data from HR sources for properties like job title and organizational division.

We employed exploratory use of the Slack workspace for some of our results. This consisted of primarily the first author using the workspace and exploring channels while taking notes of their contents. For some results, like the types of channels, an open coding scheme was used. Results stemming from exploratory use are not intended to be generalizable nor complete. Instead it is aimed to demonstrate some of the ways Slack can be used at scale.

The formal semi-structured interviews were each 1 hour long. The interviews looked to answer three key questions: how do employees use Slack during the workday; how do employees find information on Slack; and, how do employees keep up to date with what is happening at BigCorp (not necessarily using Slack). We analysed these interviews by transcribing them and finding interesting and recurring themes throughout the interviews.

#### 3.1.1 Data Anonymization

Given the private nature of communication tools we took several measures to preserve employee privacy. We only analysed "public" data that all employees at BigCorp have access to. We also did not analyse archived channels because people may "archive" a channel in an attempt to "delete" it. Additionally, we have anonymized all employees in the paper and accompanying material by blurring profile images and replacing all names with generated pseudonyms. We carefully examined each message before allowing it to appear in the paper or accompanying video. Anytime a channel name would be easily identifiable outside the company we replace it with a vegetable or fruit name and wrap it in angle brackets (eg. <eggplant>).

### 3.2 Origins of Slack at BigCorp

BigCorp has been running a company wide Slack workspace since May 2016 (3.5 years at the time of data analysis). Before that date, there were over 50 fragmented Slack workspaces being maintained by individual teams. Responding to the grass-roots demand for Slack, BigCorp decided to officially support Slack and consolidated all activity into a single shared Slack workspace and encouraged its use as an "approved" communication mechanism.

#### 3.2.1 Public by Default

The vice president at BigCorp who "sponsored" (paid for) the initial consolidation of Slack activity under a single Slack workspace agreed to do so under the condition that it would be run with a "public by default" policy, that is, that all channels would be set to "public", so they could be viewed and joined by everyone in the company – not just those in a particular team. The hope with this policy was that it would encourage openness and collaboration across the company and break down historic organizational "silos" where communication between separate teams and divisions had been limited. (*Note: "public" channels are still only visible to employees of BigCorp, not the "general public"*)

Though the policy dictates that all channels are open to everyone by default, channels do still have "members" who are subscribed to a channel. For example, a channel for a specific small development team might (and often does) only have members from that team. This can lead to the feeling that a channel is in fact private, even though it is public and anyone in the company could potentially find it. Our raw data analysis identified numerous cases of people using language inappropriate for a corporate environment, suggesting some users might not appreciate how "open" their communication on these Slack channels is. However, participant P3 was well aware of the privacy of their messages and stated that: "I'll use [private messages] … because there's just some things that don't need a public forum."

### 3.3 Analysis of Slack Usage Data

At the time of writing the Slack workspace has a total of 12,084 members and 8,204 *public* channels. The number of public channels has nearly doubled over the past year (Figure 2**Error! Reference source not found.**). There are also a small number of private channels, limited to channels where confidential HR discussions occur. We do not analyse private channels to respect employee privacy.

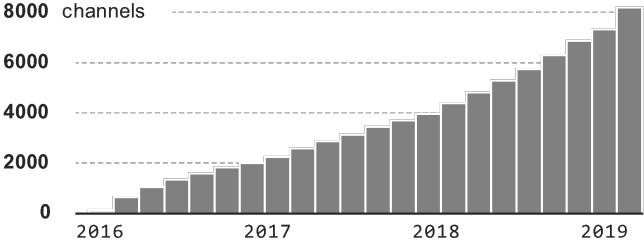

Figure 2: Number of channels over time.

In total 82 million messages have been sent in the Slack workspace with 88% of those shared as private direct messages, 11% in public channels, and 1% in private channels. Users are able to use the "direct messaging" functionality of Slack to have private conversations between 2 to 8 people. The relatively high percentage of Slack activity occurring via direct messaging is from the combined effect of people using Slack as a one-to-one instant messaging tool, and people creating "private DM groups" to essentially circumvent the mandate that all channels be public.

Despite the volume of private direct messages, the 11% of messages being sent in public channels means there have been over 9 million Slack public messages sent which are visible and searchable to all BigCorp workers, serving as a potentially rich source of company information. With such a large number of users and channels, the usage patterns among them is considerably varied.

#### 3.3.1 Channels

In addition to the 8,204 active public channels, there are an additional 6,891 archived public channels whose content (over 3.3 million messages) is still accessible through Slack search. Of the un-archived and technically "active" channels, their level of activity varies greatly with the least active 20% of channels generating less than one post per month, while the 90% percentile channel is generating 6.8 messages per month. Further, the most active channels are generating over 200 messages per month (Figure 3).

Channel membership counts are also widely distributed with half of all channels having less than 13 members, while the 40 most popular channels have over 500 members each, including the three channels to which all employees are automatically subscribed **Error! Reference source not found.**Figure 4).

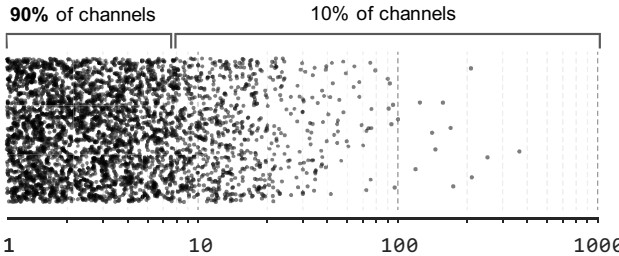

Figure 3: The number of messages posted per week, per channel. (Each dot represents one channel.)

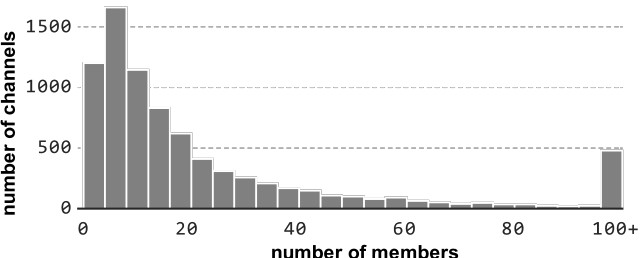

Figure 4: Channel membership distributions.

### 3.3.2 Users

The 12,084 members of the Slack workspace represents nearly every worker type at BigCorp, ranging from the CEO and VPs, to temporary contractors and outside collaborators (with limited access). Unsurprisingly, usage patterns vary among members. There are 9,417 weekly active members (have read at least one public channel in the past week), and 7,973 members who have posted at least one message in the past week. On the high-usage side, an average of 77 users post more than 100 messages in a week, and the most prolific members posting more than 400 messages in a week (Figure 5, horizontal axis).

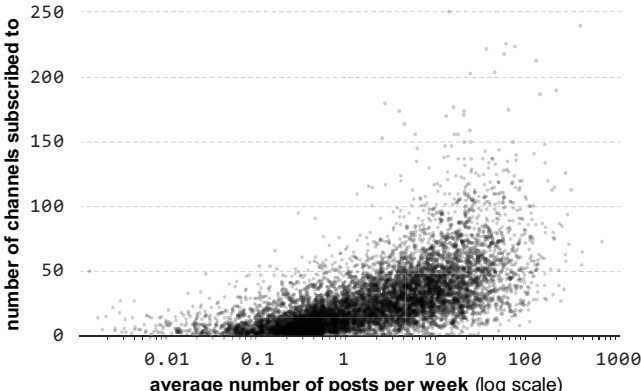

Figure 5: Plot of the number of channels subscribed to vs. the average number of messages posted each week. (Note: each dot represents a single user.)

We also see a wide range in how members subscribe to channels (Figure 5, vertical axis), with the median user subscribed to 16 channels. However, there are 168 users subscribed to over 100 channels, and 7 users subscribed to more than 200 channels.

For comparison, when using the native Slack client on a 1920x1080 resolution display, only 17 channel names will fit before scrolling is required.

### 3.3.3 Reactions

A distinguishing feature of Slack compared to more "traditional" or formal means of communication in corporate environments is the use of reactions to posts. Reactions are a quick way to respond to a message in Slack and take the form of a small emoji-like glyph or animation (Figure 6). In the corpus of public messages, a total of 422,094 reactions have been left on 220,032 unique messages. Of all reactions the :+1: "thumbs up" emoji (👍) is the most frequent with over 166,000 uses (39% of all reactions), while the popular party parrot (🦜) has been used nearly 22,000 times (5% of all reactions). Each of the 2,774 reactions used are shown in Figure 6, with their area scaled proportional to their relative usage rates.

## 3.4 Emergent Behaviour

During our exploration of the Slack workspace we discovered some interesting behaviours which had emerged, in part to facilitate using Slack at this scale.

### 3.4.1 Naming scheme

The volunteer Slack administration team has instituted a fairly strict naming scheme. Words are separated by a dash (-) and should be as short as possible. Non-business channels are prefixed with #fun- and technology channels such as #tech-git are prefixed with #tech-. Most other channels are prefixed by their organizational unit, such as #research- or #hr-.

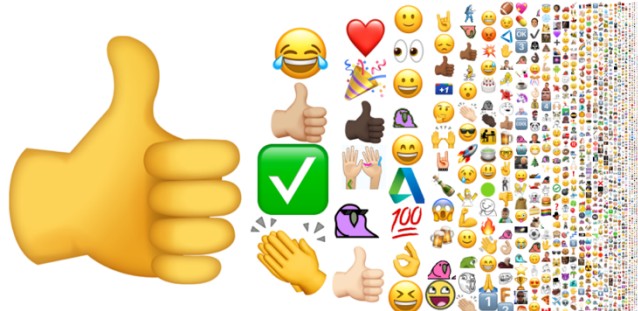

Figure 6: Emoji-style reactions used on messages in the Slack workspace, scaled by area proportional to usage.

### 3.4.2 Types of Channels

Channels naturally fell into a series of groupings. We determined the groupings by using an open coding scheme during our exploratory use of the Slack workspace.

Q&A channels are very structured channels for providing help. These channels involve a specific format for sending every message and strictly follow rules for creating threads.

News channels typically contain a collection of links and other small resources, and they often receive little engagement.

Management Team channels are composed of the direct reports of a manager and contain messages such as an employee declaring they are working from home.

Project Team channels are used to collect people on a specific project, who may come from different divisions and management chains.

Issue/Event channels are a limited lifetime channel spun up to collect communication about one specific issue. Their name corresponds directly to the issue id in the bug management system. These channels have rapid, bursty communication and fall

into disuse after the issue is resolved (typically in a matter of days).

### 3.4.3 Reactions

Several norms were formed around the way reactions were used within channels. For instance, in Q&A channels, the organizers often use eyes (👀) to indicate that a request is being addressed (essentially assigning the task). After a request is complete the message will be marked with a checkmark (✅) (Figure 7**Error! Reference source not found.**).

### 3.4.4 Events

One interesting and effective event that occurs are ask me anything (AMAs) sessions. In these events, upper management blocks off a portion of their time for employees to ask them any questions they would like. Employees engage strongly with these events and even read AMAs from divisions they are not associated with. Some of our interview participants (P2, P3) reported being very interested in hearing what upper management had to say. Other events like product launches also occur on Slack.

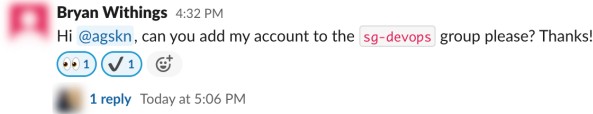

Figure 7: Asking a question in a help-specific channel with reactions.

### 3.4.5 Email vs Slack

Although a large proportion of users engage regularly with Slack, there is still a population of people who continue to use email, and even the employees who primarily use Slack still revert to email. Anecdotally, we have found that Slack is often more quickly adopted by newer/younger employees. P1 uses email "when I want to send something a bit more official". P4 preferred email almost exclusively because it is "Easier to search for things". Interview participants also reported using email to communicate with people higher up in the management hierarchy. In fact, we noticed a trend in the data suggesting that upper-management level employees tend to use Slack less often than the rank-and-file. Although there is a need for email, P3 prefers Slack because "email threads [become], you know, untenable, they would grow and they would fork." Both Slack as well as email have their own niche in which they are useful.

### 3.5 Primary Problems

Our case study identified many interesting types of Slack usage in BigCorp; however, it also identified many problems employees encounter in day-to-day use. Two problems were particularly common throughout the case study: finding historical information, and keeping up with all the activity happening on Slack. These issues align strongly with Zhang et. al's [10] findings that looked at smaller Slack workspaces.

### 3.5.1 Finding Historical Information

The first common theme throughout the case study was users having difficulty finding historical information in the Slack workspace (P1, P2, P3). So even though the workspace contains lots of potentially useful information, it is not easily accessible. P1 specifically mentioned "I find the search tool in Slack a bit cumbersome, I usually avoid having to use it" and "I might search in Slack [to find information]…, but usually that is not very productive." It is fairly unsurprising that finding historical information is difficult given the millions of messages and thousands of channels. When the huge quantity and timeline of messages is taken into account, it becomes obvious that an interface that only allows you to see one channel's messages, and only a small sample of messages at one time, will make finding historical information difficult.

### 3.5.2 Keeping up with the Workspace

The second especially common problem was with people trying to "keep up" with everything happening in the workspace. With so many channels and messages, it can understandably be overwhelming. P1 and P2 both subscribe to a large number of channels, but then rarely check them – leading to hundreds of unread messages. We found that people liked to subscribe to many channels for topics they were interested in but could not practically read all of the messages posted there. A related problem is trying to "catch up" with your channels after a vacation – with some people opting to simply "mark all messages as read" rather than trying to read or skim what they missed.

## 4 SLACKTIVITY

Slack's focus is to provide a communication tool; however, our case study shows that users need more than just communication. Employees need to be able to access historical information and also keep up with their organization. We designed a system called Slacktivity to augment the use of Slack without hindering the ability to use Slack as a communication tool.

### 4.1 Design Goals

The design of Slacktivity was guided by a set of five design goals.

*D1: Maintain Consistent Interface Vocabulary with Slack*
Our system should share the same interface vocabulary already used in Slack. By reusing concepts users are familiar with, they can transfer knowledge they already have to learn our system faster and more naturally. We found this principle especially important when sorting information. We decided on this design goal because it is important users can reuse the information they already know from Slack and are not confused.

*D2: Encourage Exploration*
The linear, time-centric nature of Slack makes exploration difficult. Our system should support exploration by breaking out of the linear consumption of messages without breaking time continuity. Encouraging exploration also stands to improve finding historical information, as some information such as entire topics or unknown query terms are not directly searchable.

*D3: Enable and Support Emergent Behaviours*
Emergent behaviour is difficult to predict and as such it is impossible to design directly for it. However, our system should enable and support emergent behaviour by allowing for flexibility. By supporting emergent behaviour, we hope to enable events like AMA's in addition to other behaviour like the use of reactions for specific purposes.

*D4: Enable Consuming Information at Varying Detail Levels*
Slack's goal as a communication tool necessitates a focus around individual messages, however users interact and visualize Slack at a higher level when they are seeking information. As a result, our system should support consumption at varying degrees of detail including those not explicitly part of Slack. Supporting varying levels of detail can also assist users in keeping up with the workspace because they do not have to look at individual messages.

*D5: Take Advantage of Organizational Data*

Organizations have other sources of data, such as HR databases. Wherever possible, we aim to take advantage of that information to help contextualize the information found within Slack within these other, existing, sources of data.

## 4.2 Implementation

The system is implemented as a web application using React combined with D3.js. Because the dataset is so large, we serve aggregate data using a Node.js server from a MongoDB.

## 4.3 Galaxy View

The *galaxy view* is the home page of the visualization, and one of the two main screens (Figure 8). It consists of four sections: clusters, trending, stream, and search. The galaxy view is designed to support exploration of the Slack workspace, as well as keeping up with the rest of the organization. Additionally, it acts as a jumping off point for accessing the messages view, the other main screen.

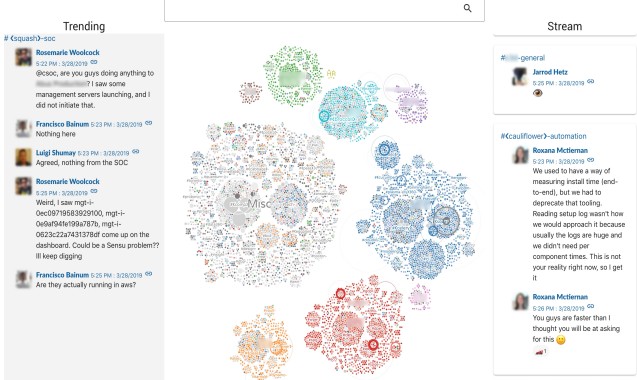

Figure 8: Galaxy view of Slacktivity.

### 4.3.1 Clusters (D4)

In the center of the screen are a series of hierarchical clusters (Figure 1). There are three concepts when describing the visualization to keep in mind: channel, prefix groups, and divisions. We will describe the clustering from the bottom upwards starting at the channels. Each circle is a channel. The radius of the circle is proportional to the square root of the number of members in the channel. The colour of the circle is determined by the organization from which the employees in that channel are a part of. If there is no majority (<50% of workers from a single organization), then the circle is coloured grey. The data regarding employee organization is obtained from an HR data source (D5), as Slack does not provide it. Additionally, the saturation of each circle is determined by the magnitude of the employee composition. For example, a pure dark blue circle will consist almost entirely of employees from the blue division.

The next level of clustering, prefixes, is formed by the prefix of the channel. shows the **#spg-** group of channels. Large channel prefixes are identified by their label, but smaller prefixes are unlabelled to improve readability.

There is one final level of clustering, divisions. This final level clusters the prefixes by the majority of the channel colours in a prefix. For example, the **#tech-** channels are mostly grey, therefore that group of channels appears with other Misc channels. This last level of clustering is meant to represent the different divisions in the organization.

The visualization is generated using a circle packing algorithm from D3.js, with the hierarchy levels described here. It also supports zooming to allow users to more carefully select a channel

they might be interested in, and to explore some of the smaller channels (D2).

The galaxy visualization reveals interesting features about the organization. For example, the green channels consist of support teams. This explains why there are often green channels inside of other divisions. These channels bridge the gap between support and product teams. Additionally, although most of the grey channels inside of "Misc" such as **#tech-** or **#fun-** channels are expressly created for all employees, there are some groups of channels that represent a product of BigCorp, like **#<eggplant>-**. These products are inter-division products that combine employees from across BigCorp, hence why they are classified as Misc.

### 4.3.2 Thumbnails (D2, D4)

Upon hovering over a channel in the cluster visualization a thumbnail will appear displaying useful information (Figure 9). It shows a line graph with the number of messages sent since the Slack workspace was created, the channel name, number of members, the total number of messages, and the top reactions sent in this channel. A user can use these features to easily determine if they would like to join a channel or explore it further. It provides information about how casual the channel is (eg. if the reaction contains a party parrot the channel is likely to be casual), how active the channel is, and how many people a user will need to interact with. Thumbnails encourage exploration by allowing users to interactively investigate channels they might find interesting. Furthermore, it provides a level of detail that Slack does not by showing the lifetime of a channel.

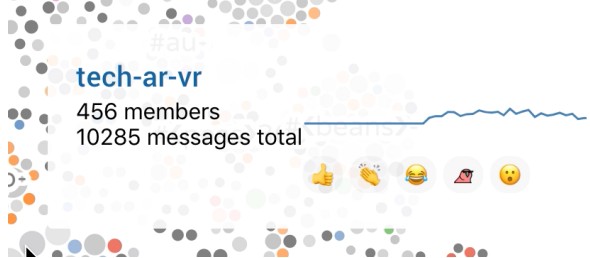

Figure 9: Channel thumbnail, the channel is listed across the top, the current number of members, total number of messages, a small trend line indicating messages sent over time, and the top 5 used reactions.

### 4.3.3 Stream

On the right side of the screen is a stream of all messages as they come into the Slack workspace (Figure 8). This is a livestream of all messages; no messages are filtered. However, they are grouped into their respective channels with the previous messages kept for context. As a message comes into the stream it is also highlighted in the galaxy view with a sonar effect that slowly dissipates as the message grows older. Some sonar effects are seen in the blue division. The clustering design of the visualization gives a few benefits can be used to see which parts of the organization are more active than others. By seeing which parts of the company are active an employee can keep up with the news of the company.

### 4.3.4 Trending

On the left side of the screen, a single trending channel is chosen to be displayed (Figure 8, left). We compute the importance score for each message and take the sum of the importance of all messages sent to a channel weighted inversely by the seconds since it was sent. This means recent messages will have a higher weighting. For each individual message we calculate four metrics: the length of the message in characters (capped at 80), the total

number of replies the message has received, the total number of reactions, the number of unique reactions, and the number of attachments the message has. Each of these values is normalised by taking the largest values and dividing them out so ] all numbers fall between 0 and 1. We then take the sum of these values.

$$importance = |replies| + min(len(msg), 80) + |reactions_{unique}| + |reactions|$$

By capturing a trending channel, we give employees the ability to keep up with the company, including paying attention to AMA's as they happen.

### 4.3.5 Search (D2)

The user can also search for a channel name by typing into the text box across the top of the screen (Figure 8). The search results contain the same visualization as the thumbnails used in the cluster visualization (Figure 9). Users can search to discover new channels they might not already be a part of as well as explore the Slack workspace as a whole.

## 4.4 Messages View

While the galaxy view gives an overview of the Slack workspace, the messages view (Figure 10), is designed to support detailed exploration and finding historical information. The messages view can be accessed by either searching for a channel from the galaxy view or clicking on a channel in the cluster visualization.

### 4.4.1 Channel Overview

Across the top of the page is the channel overview (Figure 10A). This includes the channel name, topic, and description. Additionally, there is a deep link beside the channel name to go directly into the Slack application to that channel (D1).

### 4.4.2 All Message Visualization

The primary view of this visualisation is the all message visualisation (Figure 10C). It represents all of the messages visible in the current time slice. Each message is represented as a horizontal bar, where the colour represents the user who sent the message and the length of the bar is proportional to the length of the message. Only the 5 users who have sent the most messages across the history of the channel are coloured to reduce visual clutter. Threaded replies appear as smaller boxes underneath the message they belong to. Each column of messages represents either a week, or month depending on the period of time that is currently selected. The y-vertical position of each message is decided by simply stacking bars upwards until all of the messages have been stacked. The height of each set of messages therefore represents how active the channel was for that time bucket. Hovering over a message will display a thumbnail showing the message and any threaded replies it has.

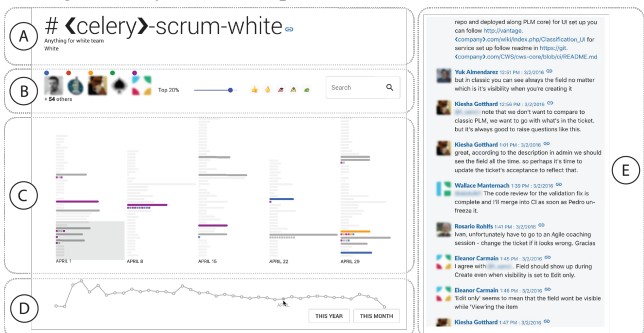

Figure 10: Messages view for a single channel with sections for A) channel overview, B) filters, C) all message visualization, D) timeslice, and E) message pane.

### 4.4.3 Message Pane (D1)

Along the right side is the messages pane which is designed to anchor the user back into the Slack workspace (Figure 10E). It is sorted reverse chronologically just like Slack, with newest messages appearing at the bottom. Each message is deep linked back into Slack. There is a light grey viewfinder in the channel visualization to represent the scroll position of where the message pane currently is. Clicking on a message in the visualization will scroll to that message in the message pane.

### 4.4.4 Timeslice (D4)

The timeslice is chosen using the timeline across the bottom of the page (Figure 10D). The line graph displays how many messages were sent during that month. Each point in the graph is a month, with the y-axis representing the total number of messages sent that month. There are two preset buttons for choosing "this month" as well as "this year". The timeslice also limits the number of messages displayed in the message pane.

### 4.4.5 Filtering

All filtering is done by the filter bar just below the channel overview (Figure 10B and Figure 11). Filters include message sender, important messages, reaction, and exact text matching by message. When a message is filtered out it is removed from the messages pane and also given reduced opacity in the visualization. Clicking a user profile image filters to only show messages from that user. This filter also acts as a legend.

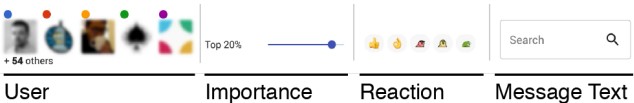

Figure 11: Filters for the messages view, allowing a user to explore a Slack channel by reducing the number of messages they need to read.

The slider filters messages using the same importance score used to detect trending. It works by adjusting the slider to control a percentile of the messages displayed. This allows users to quickly scan and review a channel by adjusting the slider to condense a channel and read only the most relevant messages.

Users can also filter by the 5 most common reactions used in the channel. Clicking one of the reactions filters to display only messages with that reaction. Supporting reaction filters allows for emergent behaviour by giving flexible filters (D3). Filtering by exact text matching is done by typing into the search bar on the far right side of the filters.

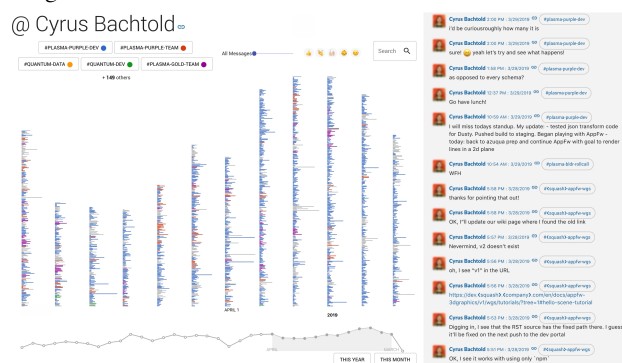

Figure 12: The messages view displaying a person rather than a channel.

### 4.4.6 User Messages View

Although we have presented the messages view as built for channels, it is also possible to invert the relationship so that the view displays a user's messages rather than a channel's messages (Figure 12**Error! Reference source not found.**). Oftentimes when searching for information a person is more important than content [8]. By allowing exploration of messages sent by a user we enable exploration of a user (D2, D4).

The user messages view can be entered by clicking on a username or profile picture from any message in Slacktivity, for example the message stream from the galaxy view or the messages pane in the messages view.

This view differs from the messages view of a channel only in that the colours in the all message visualization reflect the channel a message was sent in rather than the user who sent it. Additionally, the filters support filtering by channel rather than the message sender.

## 4.5 Example Tasks

Slacktivity can be used in several ways, but we present three sample walkthrough tasks to illustrate some useful tasks.

### 4.5.1 Task 1: A New Employee Joins AI Research

Often people transfer teams or join the company as new employees. New and transferring employees have many questions they need to answer, such as understanding how their new team functions, and discovering how their team interacts with the rest of the organization. The galaxy view can give the new employee a broad overview of the scale of the company they have joined, and how big of a niche their team is in. They can also easily compare how their coworkers use Slack by watching for activity from the prefixes their team channel belongs to and contrasting that with other divisions and prefixes.

### 4.5.2 Task 2: Exploring a Topic Across the Organization

Another employee might be interested in exploring work BigCorp has done with a particular machine learning technique. The employee might know there is a research division that researches artificial intelligence, so they begin at the galaxy view by searching for channels that begin with #research. Upon seeing an active channel, they can navigate to the messages view for that channel. The channel has thousands of messages and they do not have time to scroll through them all, so they move the importance slider to show only the top 5% of messages in the channel. From here they scroll the messages pane and read the full text of the remaining messages. Finally, they notice that about half of the important messages were sent by a single person. From this exploration the employee learned about a new channel they could join, found a knowledgeable person in the organization to ask further questions, and learned specific details from reading important chat messages.

### 4.5.3 Task 3: Keeping Up with the Organization

Rather than actively exploring Slacktivity, a user can keep a window or tab open in the background to the galaxy view. The user can then go back to the tab occasionally throughout their day and read only the trending channel. Through this, the user can discover if there is an ongoing AMA, or important discussion they can either follow or take part in. They might also see new channels they are interested in. This changes the paradigm of Slack from push communication to pull, allowing a user to keep up with the company by only spending the amount of time they would like.

## 5 DISCUSSION

In this paper we described a case study of Slack use at BigCorp. Our case study identified several interesting behaviours and pain points with Slack use at scale. To address this, we designed Slacktivity to augment Slack at BigCorp. However, there are still interesting implications and unanswered questions regarding both our case study and Slacktivity, such as generalisability and evaluation.

### 5.1 Increased Slack Use

We hope that Slacktivity helps employees get value out of using Slack. However, this might have the indirect effect of increasing the amount of time employees spend on Slack. Although increased group chat use might yield many benefits for communicating in the workplace, encouraging its use could have drawbacks. In a study on instant messaging use, Cameron and Webster [29] reported that the use of instant messaging results in more interruptions throughout the day. It is unclear whether maximising the effectiveness of Slack would produce a net benefit when contrasted with increased use.

### 5.2 Slacktivity's Effect on Privacy

Slacktivity increases the number of ways an employee can view and discover a message, essentially making all messages a little bit less private. For example, the stream in the galaxy view has the added effect of showing employees how public their messages really are. Our case study revealed that some employees might not understand how private their messages are. Depending on the perspective, helping employees understand how public their messages are could be seen as either a positive or a negative. It is negative from an organizational perspective because users might be discouraged from using public channels. BigCorp explicitly wants employees to participate in public channels to encourage an environment of collaboration and openness. However, from an employee's perspective having a realistic view of how private they are is important to maintain trust in the organization.

### 5.3 Scaling Up

BigCorp is a large organization, having more than 10,000 employees. However, there exist even larger organizations with over 100,000 employees. It is unclear whether our system scales to such a large organization, or if an organization that large would use Slack in a similar fashion. For example, if the ratio between channels (8,204) and users (12,084) remains constant then one can expect a workspace with 100,000 employees to have 67,891 channels. It seems likely so many channels require another form of organization, maybe by adding another level of hierarchy to organize channels on top of divisions and prefixes.

We suspect it is unlikely that the median channel gets larger as an organization gets larger. Project and team channels are probably the same size whether an organization has 1000 employees or 100,000 employees. However, some channels are necessarily company wide. These channels already suffer because of the huge population that messages to them address. Sometimes mistakes happen and somebody notifies the entire channel by erroneously using @channel. This problem might be exacerbated when even more employees are in a channel.

### 5.4 Scaling Down

On the opposite end of the spectrum it is interesting to consider whether our case study and Slacktivity also generalise to small businesses with 100 or even medium businesses with 1,000 employees. The messages view probably has the same value because it was designed to work with small channels as well as with massive channels. Most of our findings regarding how Slack

is used probably scale to medium sized organizations, as managing that many channels will require strict naming schemes and other forms of organization.

## 5.5 Generalisability

We have presented a case study of the use of Slack in a large organization. The question stands: do our results generalise to other organizations? This is a difficult question to answer because most organizations are unwilling to publicly disclose the detailed information described in our case study. However, we have some confidence that some aspects do generalise. 65 of the Fortune 100 companies use Slack, indicating that other companies do use Slack at a large scale. Other large organizations like IBM have channels "public by default" [30]. Not only do some of the more critical components of our case study generalise, but other parts such as issue specific channels are used in companies like Intuit [31] and other companies host events using Slack [32], [33]. It stands to reason that many of the other problems and descriptions of Slack use at scale described in this paper generalise to other large organizations.

A limitation of our work is that we do not evaluate Slacktivity with a user study. However, our example walkthroughs make an argument as to how Slacktivity might be used to solve some of the problems in our case study. Furthermore, this paper introduced novel methods of visualising Slack at scale and a detailed case study of how Slack is used at scale today. Future iterations of this work will be deployed at BigCorp, allowing us to study the long term impacts a deployment of Slacktivity might have.

## 6 CONCLUSION

In this paper we presented a case study of how Slack is used at BigCorp, a large organization with over 10,000 employees. Our case study highlighted interesting behaviour of Slack usage at a large corporation, and problems encountered by employees using Slack. Based on the case study we introduced a novel visualization technique, Slacktivity, designed to augment Slack in the workplace and enable its use at scale.

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
