# OpenReview forum: "Slacktivity: Scaling Slack for Large Organizations"
_graphicsinterface.org/Graphics_Interface/2021/Conference/Second_Cycle — Reject_

### Official Review · Reviewer_mo3V · 2021-04-28
**This paper presents the results of a mixed methods study of barriers and practices of Slack use in large organizations. It contributes Slacktivity - a visualization tool to help users overcome the barriers of using Slack at scale.**

**Rating:** 4
**Confidence:** 4

**Review:**

OVERALL COMMENTS
This reviewer found that the abstract was clearly written and that it explores an interesting topic - how people manage the use of Slack at scale. Overall however, this paper suffered from a lack of cohesive narrative on a rigorous and logical methodology. The major weakness is the logistical framing and flow of the paper --- the intro, related work, user study, system design all seemed to be disjoint from each other. There seems to be a promising line of work here - I encourage the authors to improve the narrative and framing of the methodology of this manuscript and resubmit it once improved, at a later point in time.

LOGISTICAL FLOW OF THE PAPER
This reviewer found the flow of the paper quite strange. The links between intro, related work, user study, study findings, system design, example tasks are disjoint. Specifically:

In the related work section, it doesn't make sense that visualization tools are already suggested as a solution to improve barriers to using Slack at scale. If visualization is already suggested as a potential solution, it is unclear why the authors also chose to undertake a user study in Section 3.  Was the user study actually necessary? If so, it was not clearly justified.

Furthermore, Section 3.4 presents the analysis of the user study, while Section 3.5 identifies some challenges that users have experienced.   This is fine except that Section 4.1 (Design goals) don't seem at all grounded in the findings from the user study or grounded in related work. At present, design goals seem like it came from the authors in an impromptu brainstorming session. How did you come up with these design goals? How are they related to the findings from the user study?

Next, Section 4 presents the design and implementation of Slacktivity. This section has some logical weaknesses. First, the design of Slacktivity is not actually grounded to the interview findings. Second, as no user study was conducted to evaluate the impacts of Slacktivity, it is unclear if Slacktivity would actually help users overcome or mitigate the barriers identified previously.

Instead, Section 4.5 (Example tasks) present some user scenarios, though it is unclear how these scenarios were generated, and whether they are realistic or not. At present, there are a lot of implicit assumptions in the example tasks, about how users would use Slacktivity. Since no evaluation of Slacktivity was conducted, it is unclear how much value these example tasks offer.

INTRODUCTION
The motivation and framing of the paper in the introduction could be punchier. How many companies use Slack? Is using Slack (or related tools) at scale a common phenomenon? Do businesses typically wish to use Slack to derive high level or historical insights? Adding some statistics like this could strengthen the intro.

RELATED WORK
Depending on how the authors re-frame this paper in future iterations, they may wish to distinguish whether the design of Slacktivity is a novel visualization, or an already existing visualization, applied to a novel context.

FIGURES
All figure-related text is missing in the paper. (says "error reference source not found").

---

### Official Review · Reviewer_U3mW · 2021-05-03
**Good application of HCI methods, unclear research contribution**

**Rating:** 5
**Confidence:** 4

**Review:**

This paper describes an interface (Slacktivity) to visually explore large slack workspaces. Among the things to like in this paper are:
- it is a (relatively) well-written manuscript,
- the case study uses real data from a large corporation,
- the strategies that people employ to deal with such large slack workspaces, as identified by the authors, is a valuable contribution,
- the system is well designed.

One of the key aspects of the proposed system that differs from previous work is that the focus is more on the relationship between items than on the linear sequences of items.
The system itself does not make a visualization contribution.

The methodology, relying on multiple data sources, and providing a mix of qualitative and quantitative information on how slack is being used at <BigCorp>, is very appropriate. It provides a good overview of how a tool like Slack is used in a large corporation, and a bit of the why as well (5 interviews is not a lot, yet it still provides some interesting qualitative insights). It is a bit unclear how the qualitative data was collected, formatted, and analyzed, and this should be explained in greater details.

I have some trouble relating the design goals to the data analysis in the previous section. Some links are more or less clear, like D1 that can be linked to 3.4.1 - Naming scheme, however, this would be much easier to follow and more convincing if these links were made explicit.
While the system description is quite clear and the design decisions make sense, it is unclear how these were devised. Some of the design choices and features are linked to the design goals (e.g., 4.3.1, 4.3.2, 4.3.5), but others are not (e.g., 4.3.3, 4.4.5, 4.4.6). In the end, it does not convince me that the design decisions were made based on the data collected and analyzed, but rather that the design goals were devised once the system had been designed. This is something that should hopefully be fixed with some rewriting.

The example tasks are fairly simple scenarios that do not help at evaluating the usefulness of the proposed system.

Overall, this is the second time I review this paper, and unfortunately it did not improve much, and many of the (good) comments made by previous reviewers have not been addressed.

I wrote earlier that it is a (relatively) well-written manuscript, because although it is overall easy to read and follows a clear structure, there is also a real sloppiness in the work, as illustrated by the following typos and presentation issues.
- associated with group. (p2)
- (Figure 2Error! Reference source not found.). (p2)
- 3.1.1 is the only sub-sub-section in 3.1.
- automatically subscribed Error! Reference source not found.Figure 4 (p4)
- The 12,084 members of the Slack workspace represents (p4)
- Figure 7Error! Reference source not found. (p5)
- Figure 12Error! Reference source not found.) (p8)

---

### Official Review · Reviewer_aXHj · 2021-05-04
**Good pick and framing of the research direction, but the case study and visualization design can be better connected and integrated.**

**Rating:** 5
**Confidence:** 4

**Review:**

The paper targets at the phenomenon of large-scale usage of Slack in organizations by presenting a case analysis of corporate usage of Slack among 10,000 of employees, and then proposing a visualization system Slacktivity with the goal to support the use of Slack in large organizations to overcome observed challenges in exploring the content and keeping up with news across different channels of the company.

Overall, I found the general direction of research and design interesting. It is quite insightful to note that group chat programs like Slack can provide the important function of knowledge management, acting as a "knowledge base" for a company, which has motivated the approach to develop a visualization system to visualize historical chat data quite well. The introduction of the paper is really nicely written, which has really shown a promise for the case study and the design work to thrive in the outlined problem context. The dataset being analyzed in the case study is impressively large, with more than 10,000 members, 8000+ channels and millions of messages. The data analysis is complemented with a qualitative interview involving 5 employees. The paper is also overall well-written and easy to follow.

While the entire set-up of the study and design appears to be very interesting and noteworthy, the case study, the visualization design and most importantly, the connection and integration between the study and the design all appear to have some space of improvement, rendering the paper a hint of incompleteness of the paper.

For the study part, the  analysis of the slack usage data, unfortunately, didn't really impress us with an in-depth understanding. The analysis did provide a snapshot of how the corporate Slack space looks like with distributions of channels, membership and messages posted. There're also some interesting, yet nuanced observations of emergent behaviors, such as the naming convention, the use of emojis, comparison of emails and group chat, and question asking on the chat etc. While I appreciate to have access to the close-up picture of the corporate slack as a background of the work, I didn't quite see how the descriptive statistics presented help us to obtain in-depth insights/implications for design. It's also unclear what research questions the data analytics help to answer. While there's a very good motivation behind this work, the research questions to answer weren't explicit in the paper.

The 5-person qualitative interviews did provide useful information regarding the difficulty to find historical information from the channels, and the need to keep up with the corporate workspace at this overwhelming scale. But the qualitative part of the study is relatively brief and underdeveloped. It also feels disconnected with the quantitative part of the case study. It can be much more persuasive if the different methods work in conjunction to answer the same set of research questions empirically, and generate a coherent piece of understanding on what have been challenging (and in what way it's challenging) to users of the slack channels in the corporate context.

For the design part, the design goals are not entirely informed or driven by findings from the study. The visualization and interface design look nice and reasonable to me. I particularly liked the design of the "galaxy view" of the system, which is not just aesthetically pleasing, but could also potentially improve the usability and accessibility when exploring and consuming slack messages. One thing unclear to me is that the current design seems to support only content consumption (e.g., reading and exploring messages, mostly historical), but not content production (e.g., composing and participating in discussions, which can be historical and real-time).  I wonder whether supporting consumption along is sufficient for making Slack more scalable and accessible in the company?

There's lack of evaluation of the system, which is certainly a weakness of the paper. While evaluation isn't always a requirement for technical work, given the purpose of this paper, evaluation is still essential to provide a proof-of-concept for the system proposal.

---

### Meta-Review · Area_Chair_jfEa · 2021-05-05

**Recommendation:** Reject
**Confidence:** 5

**Metareview:**

This paper contributes a case study of Slack usage at a large organization and a visualization (Slacktivity) to reveal patterns of how Slack is used at scale. Reviewers found the overall framing and direction of this paper interesting and compelling, and were impressed by the access to a large dataset of Slack usage at a real organization. However, all reviewers found that sections of the paper were disjoint (research questions, study design, data collected, visualization design, example tasks) and had concerns regarding the methodology. Consequently, the significance and contribution of the paper were unclear. One reviewer is providing feedback on this manuscript for the second time, and noted that previous concerns from reviewers were not addressed in this submission. There are also obvious grammar and spelling issues throughout the work, which gives the impression of lacking attention to detail. Overall, I recommend rejection and strongly encourage the authors to address the thoughtful and detailed feedback given by reviewers. In future iterations of this work, the authors may consider making major changes to re-frame this paper using a more cohesive narrative with regards to the research questions, methodology and significance of contribution.

---

### Decision · Program_Chairs · 2021-05-08

Reject